# Identification of Bioactive Phytochemicals from Six Plants: Mechanistic Insights into the Inhibition of Rumen Protozoa, Ammoniagenesis, and α-Glucosidase

**DOI:** 10.3390/biology10101055

**Published:** 2021-10-18

**Authors:** Aurele Gnetegha Ayemele, Lu Ma, Xiumei Li, Peilong Yang, Jianchu Xu, Zhongtang Yu, Dengpan Bu

**Affiliations:** 1State Key Laboratory of Animal Nutrition, Institute of Animal Sciences, Chinese Academy of Agricultural Sciences, Beijing 100193, China; ayemeleaurel@yahoo.com (A.G.A.); malu@caas.cn (L.M.); 2Honghe Center for Mountain Futures, Kunming Institute of Botany, Chinese Academy of Sciences, Honghe County, Kunming 654400, China; 3Feed Research Institute, Chinese Academy of Agricultural Science, Beijing 100081, China; lixiumei@caas.cn (X.L.); yangpeilong@caas.cn (P.Y.); 4World Agroforestry Center, East and Central Asia, Kunming 650201, China; 5Department of Animal Sciences, The Ohio State University, Columbus, OH 43210, USA; yu.226@osu.edu; 6Joint Laboratory on Integrated Crop-Tree-Livestock Systems of the Chinese Academy of Agricultural Sciences (CAAS), Ethiopian Institute of Agricultural Research (EIAR) and World Agroforestry Center (ICRAF), Beijing 100193, China

**Keywords:** bioactive phytochemicals, feed efficiency, microbial α-glucosidase, protozoa inhibition

## Abstract

**Simple Summary:**

Rumen protozoa have some contribution to feed digestibility in the rumen, but *Entodinium*, the most predominant genus, is the main culprit of inefficient nitrogen utilization in ruminants. Using chemical drugs, many studies have attempted to inhibit the rumen protozoa, but few of the approaches are either effective or practical. In this study, we investigated the nutritional and functional properties of *Adansonia digitata* (baobab), *Flemingia macrophylla* (waras tree)*, Kalimeris indica* (Indian aster)*,*
*Brassica rapa* subsp. *chinensis* (bok choy), *Portulaca oleracea* (common purslane), and *Calotropis gigantea* (giant milkweed) for their potential as feed additives in animal husbandry. The plants were also analyzed for their major phytochemicals using reversed phase-high performance liquid chromatography (HPLC) and then evaluated for their ability to inhibit rumen protozoa, ammoniagenesis, and microbial α-glucosidase activity in vitro. *C. gigantea* inhibited the rumen protozoa and reduced the wasteful ammoniagenesis, thereby indicating improved nitrogen utilization. *A. digitata* also reduced the microbial α-glucosidase activity that can potentially contribute to rumen acidosis. The tested plants, especially *C. gigantea* and *A. digitata*, could be used as potential alternatives to chemicals or antibiotics to ensure sustainable and green animal husbandry.

**Abstract:**

Rumen protozoa prey on feed-degrading bacteria synthesizing microbial protein, lowering nitrogen utilization efficiency in ruminants. In this in vitro study, we evaluated six plants (*Adansonia digitata*, *Flemingia macrophylla, Kalimeris indica**,*
*Brassica rapa* subsp. *chinensis, Portulaca oleracea*, and *Calotropis gigantea*) for their potential to inhibit rumen protozoa and identified the phytochemicals potentially responsible for protozoa inhibition. Rumen protozoa were anaerobically cultured in vitro in the presence of each plant at four doses. All of the tested plants reduced total rumen protozoa (*p* ≤ 0.05), but *C. gigantea* and *B. rapa* were the most inhibitory, inhibiting rumen protozoa by 45.6 and 65.7%, respectively, at the dose of 1.1 mg/mL. Scanning electron microscopy revealed a disruption of the extracellular structure of protozoa cells. Only *C. gigantea* also decreased the wasteful ammoniagenesis (*p* ≤ 0.05). Moreover, the *A. digitata* extract inhibited α-glucosidase activity by about 70% at 100 µg/mL. Reversed-phase high-performance liquid chromatography analysis detected quercetin, anthraquinone, 3-hydroxybenzoic acid, astragaloside, and myricetin in the tested plant leaves. These plants may hold potential as feed additives to reduce rumen protozoa and α- glucosidase activity. Future research is needed to identify the specific anti-protozoal compound(s), the effects on the rumen microbiome, and its fermentation characteristics.

## 1. Introduction

Ruminants can convert fibrous plant materials into meat and milk for human consumption. However, improvement of the feed efficiency remains a big challenge, especially nitrogen utilization efficiency (NUE), which is no more than 30% as most of the dietary nitrogen is excreted as urea and ammonia [1], negatively impacting the environment and increasing feed costs. In particular, rumen protozoa engulf the cells of rumen microbes (primarily bacteria), which are the main source of metabolizable protein for the host animals, and degrade the microbial protein into oligopeptides and free amino acids, both of which are converted to ammonia, lowering NUE [2]. Some antimicrobials have been used to improve feed utilization efficiency including NUE, but unfortunately, some side effects on both animal products and human consumers have been reported [3]. This led to the ban of antimicrobials as feed additives and calls for renewed interest in using plants as a source of natural alternatives, especially in organic animal husbandry. A recent study [4] demonstrated that the leaves of *Calotropis gigantea*, which contains phytochemical groups, such as phenolics, flavonoids, and alkaloids, could effectively inhibit rumen protozoa by nearly 60%, thereby decreasing wasteful ammoniagenesis in vitro. More importantly, *C. gigantea* did not adversely affect any of the fermentation characteristics. No study, however, has described the mechanism of the inhibition of rumen protozoa by any bioactive phytochemicals. Using scanning and transmission electron microscopy, Park et al. [5] showed that antibiotics and inhibitors of protozoal digestive enzymes could destruct the cellular structure of *Entodinium* and other rumen protozoa both extracellularly and intracellularly. 

High-producing ruminants are generally fed diets rich in rapidly fermentable carbohydrates, leading to an upsurge of volatile fatty acids (VFA), and consequently a dramatic decrease in the rumen pH, which in turn causes ruminal acidosis and associated metabolic disorders and decreases milk yield [6]. The rapid degradation of carbohydrates or polysaccharides is due to the enzymatic activity of microbial amylase and glucosidase, which must be regulated to mitigate rumen acidosis. To that end, acarbose has been used as a synthetic inhibitor of α-glucosidase to prevent the rumen pH from rapidly dropping, but unfortunately, similar to other synthetic drugs, acarbose was associated with serious side complications [7]. In addition, the use of acarbose to alleviate ruminal acidosis also led to the increase in ammonia nitrogen (NH_3_–N) concentration [8], which decreases NUE. Moreover, as a drug to treat diabetes, acarbose is too expensive to be fed to ruminants. It accordingly now becomes urgent to explore natural alternatives from plants as alternatives to inhibit microbial α-amylase and α-glucosidase activity and to lower the risk of rumen acidosis. 

Natural plants are a potential source of novel biologically active compounds that could lead to metabolic interventions as new therapeutics [9,10]. The recent discovery of a powerful antimalarial drug, artemisinin, derived from *Artemisia annua* L. (sweet wormwood) [11] is recommended by the World Health Organization to cure malaria disease, caused by *Plasmodium falciparum* parasite [12]. Traditional Chinese medicine is devoted to treating a wide range of infectious diseases [13], and several phytochemical compounds are already undergoing clinical trials [14]. High throughput screening has enabled a variety of phytochemicals to be searched for their chemical, bioactivity, and ethnobotany information (https://phytochem.nal.usda.gov/phytochem/search/list; 16 August 2021) and for their potential targets of proteins and nucleic acid targets (https://db.idrblab.org/ttd/; 16 August 2021) [15]. 

Some forest plants and their fruits, such as the fruit pulp of *Ada**nsonia digitata* (baobab), have been shown to inhibit α-glucosidase activity [16]. Those plants contain various bioactive and antimicrobial phytochemicals, yet most of them have not been explored for their potential as feed or feed additives for ruminants. In a recent study [4], we found that *C. gigantea* could markedly inhibit rumen protozoa and decrease NH_3_–N production in vitro. The objective of the present study was to analyze another five plants in comparison with *C. gigantea* for their potential to inhibit rumen protozoa, to decrease ammonia production, and to regulate the activity of α-glycosidase using in vitro rumen cultures. We hypothesized that some of those plants, such as *Flemingia macrophylla* (waras tree)*, Kalimeris indica* (Indian aster)*,*
*Brassica rapa* subsp. *chinensis* (bok choy)*, Portulaca oleracea* (common purslane), and *C. gigantea*, none of which had been tested in animal husbandry, could be used as a source of protein and fiber in typical diets of ruminants and at the same time as inhibitors of detrimental protozoa, ammoniagenesis, and starch digestion. We tested this hypothesis by evaluating the above plants for their inhibitory effect on total rumen protozoa and α-glucosidase and by identifying their major phytochemicals to aim for reaching a mechanistic understanding of their holistic effects. 

## 2. Materials and Methods

### 2.1. Plants Collection and Proximate Analysis

Forest plants were collected in Honghe, located in the southwest of Yunnan Province, China. The plants were selected based on their ethnobotanical value in traditional Chinese medicine. They were taxonomically identified and confirmed by Prof. Liu Yi, a botanist of the Herbarium Botanical Institute of Kunming, Chinese Academy of Sciences, where their voucher numbers are recorded. They included *A. digitata* (FJSI011773), *B. rapa* subsp. *chinensis* (BNU0018604), *C. gigantea* (KUN0307207), *F. macrophylla* (KUN0616353), *K. indica* (KUN0983258), and *P. oleracea* (KUN1063740). After collection, the plant parts were used following standard operating procedures established by the Good Agriculture Practices [17]. In brief, the leaves were separated from the stems, sun-dried, and ground up to pass through a 2 mm mesh for nutritional analysis and a 0.3 mm mesh to be used as the feed of the protozoal cultures and subjected to crude extraction for phytochemical analysis (for *F. macrophylla*, both leaves and roots were used, and for *B. rapa chinensis*, only its tuber was used). Finally, the ground plant samples were stored in individual glass bottles and kept at room temperature until further analysis. Nutritional analysis was conducted using standard methods [18] for dry matter (DM), ash, crude protein, and ether extract. Neutral detergent fiber (NDF) and acid detergent fiber (ADF) were determined using the methods outlined by Goering and Van Soest [19]. 

### 2.2. Chemical Analysis of Plants

#### 2.2.1. Plant Sample Preparation and Extraction

The finely ground plant samples (0.3 mm particles) were individually subjected to chemical extraction according to the procedures described by Ayemele et al. [4]. In brief, 15 g of each ground plant sample was combined with 450 mL of 80% ethanol, and the mixture was subjected to ultrasonication using a Soniprep 150 Ultrasonicator (Hongxianglong Biotechnology Co., Ltd. Beijing, China) at 55 °C for 45 min at 90% of its maximum power level. The sonicated samples were then centrifuged (10,000 rpm for 10 min), and the supernatants were collected and evaporated at 55 °C in a rotary evaporator set at 85 rpm. Around 20 mL of the crude extract from each sample was freeze-dried at −70 °C for 72 h, and the final extract was weighted.

#### 2.2.2. Quantification of Total Phenolic, Flavonoid, and Alkaloid of the Crude Extract

The crude extract samples were analyzed for total phenolic compounds (TPC), total flavonoid compounds (TFC), and total alkaloid content (TAC) as previously described [4,20,21]. The content (per gram of plant leaves, DM) of TPC, TFC, and TAC were estimated as µg of gallic acid equivalent (µg GAE/g), µg of rutin equivalent (µg RE/g), and µg of aconitine equivalent (µg AE/g), respectively. 

#### 2.2.3. Reversed-Phase HPLC for Quantification of Individual Plant Metabolites 

Major secondary metabolites in the crude extract samples were analyzed using reversed-phase HPLC (RP-HPLC) as described by Ayemele et al. [4]. In brief, the analysis consisted of separation, detection, and quantification of the compounds using the Agilent ZORBAX EclipsPlus C-18 reversed-phase column, a diode array detector, and an autosampler (Agilent Technologies, 1290 Infinity II, Palo Alto, CA, USA). Spectral data were recorded from 200 to 800 nm, and the chromatograms of the compounds of interest were monitored at 326 nm. The chromatographic peaks of the plant compounds were confirmed by comparing the retention time with those of the following reference standards: 3-hydroxybenzoic acid, myricetin, astragaloside, quercetin, and anthraquinone. The content of individual compounds was expressed as µg/g of plant samples (DM).

### 2.3. In Vitro Culture of Rumen Protozoa and Evaluation of Plant Effects

#### 2.3.1. Ethics Statement

Animals were handled and cared for following the protocol approved by the Animal Care Advisory Committee of the Chinese Academy of Agricultural Sciences, Institute of Animal Sciences (Protocol No.: IAS20180115), which approved the rumen fluid collection method from the dairy cows.

#### 2.3.2. Collection of Rumen Fluid and Preparation of Concentrated Rumen Protozoa 

Rumen fluid was collected from three cannulated dairy cows fed a typical total mixed ration (TMR) 2.5 h after the morning feeding. To eliminate the exposure of anaerobic rumen microbes to the air, the bottles were filled to the rim and tightly closed. Then, the rumen samples were transported to the laboratory within 1.5 h. The rumen fluid samples were kept at 39 °C in thermally insulated bottles while being transported to the laboratory to minimize loss of microbial viability [4]. In order to limit the microbial disparity between the donated cows, an equal volume of the three rumen fluid samples was combined into one and filtered through four layers of cheesecloth into a 500 mL glass bottle under a continuous flux of COAfter filtration, the large feed particles were removed, and the protozoa cells were left to sediment under a continuous O_2_-free CO_2_ flux. The concentrated cells of the typical protozoa genera including the other microbes found in dairy cows were used as the inoculum for the in vitro culture experiments and evaluation of the plants. 

#### 2.3.3. In Vitro Culture of Rumen Protozoa with Plant Supplementation 

The in vitro culture technique of rumen protozoa was used as reported by Ayemele et al. [4]. In brief, 8 mL of medium [22] was dispensed into individual glass culture tubes containing 100 mg of the same TMR fed to the lactating cows that donated the rumen fluid. Each of the ground plant samples was added to the glass culture tubes at four doses (0, 0.7, 0.9, and 1.1 mg/mL culture fluid), with each dose having three replicates (n = 3). Finally, 2 mL of the enriched-protozoa rumen fluid was added under a continuous flux of O_2_-free CO. The tubes were incubated at 39 °C for 24 h. Each in vitro culture was sampled for protozoa counting and determination of NH_3_–N concentration. 

#### 2.3.4. Microscopic Counting of Rumen Protozoal Cells 

The protozoal cells were morphologically identified and counted microscopically using the procedures previously described [4,5]. In brief, 0.3 mL of each in vitro culture sample was combined with 0.3 mL of 18.5% formalin to fix the protozoal cells. Then, 30 µL of a brilliant green dye solution was added to stain the protozoal cells. Finally, 1.4 mL of 30% glycerol was added to each tube and mixed. The cells were counted using a Sedgewick-Rafter counting chamber (Thomas Scientific, Swedesboro, NJ, USA) under a light microscope at 10× magnification. 

#### 2.3.5. Scanning Electron Microscopy of the Protozoal Cells Surface 

The samples of cultured rumen protozoa cells were prepared for scanning electron microscopy (SEM) following the procedures previously described et al. [4]. In brief, protozoal cells were pelleted by centrifugation at 500× *g* for 5 min from 1 mL of each in vitro culture. Then, the cells were fixed with 3% glutaraldehyde and subsequently rinsed twice with a potassium phosphate buffer (0.1 M, pH 7.2). A sequential washing of protozoal cells with 30, 50, 70, 90, and 100% ethanol helped to dehydrate the cells, followed by washing in acetone and then hexamethyldisilazane. The cells were then spatter-coated with platinum and viewed on a Hitachi S-4700 (Hitachi America, Ltd., White Plains, NY, USA) after five hours of drying with a flux of CO_2_. 

### 2.4. Detection of Ammonia Nitrogen Concentration 

NH_3_–N concentrations of the culture fluid were determined using the colorimetric method described by Chaney and Marbach [23]. In brief, 500 µL and 400 µL of the solutions A and B, respectively, were successively added and mixed with 8 µL of each sample and then incubated for 30 min at 37 °C. In a 96-well microplate, 200 µL was dispensed to each well and optical absorbance was determined at 550 nm wavelength using a spectrophotometer (Thermo Fisher Scientific, Waltham, MA, USA). A serial dilution of ammonia solution (1, 2, 4, 8, 16, and 32 mg/dL) served as the external standard. 

### 2.5. α-Glucosidase Inhibition Assay 

The α-glucosidase inhibition assay was performed according to the procedures of Obaroakpo et al. [24] and Rengasamy et al. [25] with minor modification. In brief, α-glucosidase obtained from *Saccharomyces cerevisiae* was dissolved in 0.1 M potassium phosphate buffer (PBS, pH 6.8) and used as the enzyme stock solution (0.1 Unit/mL). The substrate for the enzyme activity assay was *p*-nitrophenyl-a-D-glucopyranoside (pNPG) dissolved in the same PBS buffer (stock concentration of pNPG: 0.375 mM). Inhibition in the yeast α-glucosidase was determined at five concentrations ranging from 100 to 500 µg/mL with an increment of 100 µg/mL of each plant extract in PBS, using 96-wells microplates. Acarbose, which is an inhibitor of α-glucosidase, served as the positive control for the above concentrations. Each α-glucosidase assay reaction contained 20 µL of each plant extract solution, 20 µL of α-glucosidase enzyme solution, and 40 µL of pNPG. The mixture was incubated at 37 °C for 40 min, and then 80 µL of 0.2 M sodium carbonate in PBS was added to terminate the reaction. Optical absorbance was read using a spectrophotometer (Thermo Fisher Scientific, Waltham, MA, USA) at 405 nm. One negative control that contained no plant extract or acarbose was included in parallel. Each assay reaction had three replicates. The inhibition (%) was calculated using the following equation:Inhibition = (A_control_ − A_sample_)/A_control_ × 100 (1)
where A_control_ is the absorbance of the negative control (no inhibitor) and A_sample_ is the absorbance for each plant extract. 

A nonlinear regression based on five different extract concentrations ranging from 100 to 500 µg/mL was used to calculate the concentration of each crude extract that inhibited the α-glucosidase activity by 50% under the assay conditions, defined as IC_50_ expressed in μg/mL. 

### 2.6. Statistical Data Analysis

The data were analyzed in a complete randomized design with one-way ANOVA using the PROC GLM procedure of SAS 9.4 (SAS Institute, Cary, NC, USA) to compare the means of nutrients and phytochemicals among the different plants and the protozoal cell counts among the different doses of each plant sample. The means of the α-glucosidase inhibitory activity were also compared among the different doses within each plant. Orthogonal polynomial contrast was conducted to determine the linear and/or quadratic effects of plant doses on protozoal cells and NH_3_–N concentration. Significance was declared at *p* < 0. The GC/MS Translator B.07.17 was used to convert the Chemstat HPLC data files into MassHunter files for qualitative and quantitative analysis of the compounds in the plant extracts.

## 3. Results

### 3.1. Proximate Composition and Total Polyphenols of the Analyzed Plants

Crude protein, ether extract, fiber, and ash were the main nutrients detected on a dry matter (DM) basis (Table 1). *K. indica* recorded the highest protein content, while *F. macrophylla* roots the lowest. *F. macrophylla* and *C. gigantea* had higher NDF and ADF contents than the other plants. The ether extract content was no more than 1.9% of any of the tested plants, except for *C. gigantea*, which recorded the highest value (3.9%). Ash content was found higher in *P. oleracea* and *C. gigantea* than in the other plants. The roots of *F. macrophylla* had the lowest ash content. The plants had different yields of extracts and different content of phenolics, flavonoids, and alkaloids. *B. rapa* subsp. *chinensis* (the tubers) had the highest crude extract yield, while *F. macrophylla* and *P. oleracea* recorded the lowest. The phenolic content was found to be the highest in *F. macrophylla*, followed by *A. digitata* and *C. gigantea*. *B. rapa* subsp. *chinensis* (tubers) and *P. oleracea* had the lowest phenolic contents. The highest total flavonoid content was found in *C. gigantea* and the lowest was found in *B. rapa* subsp. *chinensis* (tubers). Except for *F. macrophylla*, which had the highest total alkaloid content, the other plants recorded the lowest content of alkaloids compared with that of phenolics and flavonoids.

### 3.2. Quantification of Some Individual Phytochemical Compounds

Among the studied plants, *A. digitata* appeared rich in 3-hydroxybenzoic acid, astragaloside, and myricetin as it had the highest content of those individual compounds, while *K. indica* was rich in 3-hydroxybenzoic acid; *C. gigantea* was rich in quercetin, and *F. macrophylla*, *P. oleracea,* and *B. rapa* subsp. *chinensis* were rich in astragaloside (Table 2). The *F. macrophylla* roots also contained a high level of 3-hydroxybenzoic acid. The chemical structure of the individual compounds is shown in Figure 1.

### 3.3. Effects on Rumen Protozoal Counts and Ammoniagenesis in the In Vitro Cultures 

Decreased total protozoa counts were observed in all of the tested plants although to different extents (Figure 2). At the tested maximum dose of 1.1 mg/mL, *B. rapa* subsp. *chinensis* had a higher inhibitory effect, decreasing the protozoa counts from 140,000 to 48,000 cells/mL, while *A. digitata* and *C. gigantea* reduced protozoa from 140,000 to 73,360 and to 76,160, respectively (*p* < 0.05). *K. indica, F. macrophylla*, and *P. oleracea* reduced the protozoal count to 106,400, 126,700, and 128,800, respectively. Only *C. gigantea* and *A. digitata* decreased protozoa at the lowest dose tested (0.7 mg/mL) (*p* < 0.05). *C. gigantea* reduced (*p* < 0.05) the concentration of NH_3_–N, whereas *F. macrophylla* (both leaves and roots) increased ammoniagenesis (*p* < 0.05). *B. rapa* subsp. *chinensis**, K. indica, P. oleracea,* and *A. digitata* did not affect (*p* > 0.05) the concentration of NH_3_–N (Table 3).

### 3.4. Disruption of the Cells Surface of Entodinium by the Tested Plants 

The extracellular surface (pellicles) of *Entodinium* cells collapsed and wilted with the tested plants (Figure 3). The normal longitudinal striations present on the cell surface disappeared after the exposure to *B. rapa* subsp. *chinensis**, K. indica,* and *A. digitata,* but *C. gigantea, F. macrophylla*, *P. oleracea,* or the control did not destruct the striations.

### 3.5. Inhibition of α-Glucosidase Activity by Phytochemical Extracts

The extract of *A**. digitata* inhibited the activity of yeast α-glucosidase with 310.11 μg/mL of the extract showing 50% of inhibition of the α-glucosidase activity (IC_50_). Additionally, the *A. digitata* extract inhibited the enzyme in a dose-dependent manner while the extract of *B. rapa chinensis* did cause a stable inhibition at the tested doses, except for the dose 400 µg/mL (Figure 4). As expected, acarbose inhibited the α-glucosidase activity in a dose-dependent manner.

## 4. Discussion

Many synthetic products are still used either as inhibitors of certain gut microbes and their enzymes or growth promoters at different levels to control or regulate some physiological functions of food-producing animals. Unfortunately, those synthetic products are nowadays recognized to have adverse effects associated with global public health concerns. Therefore, many of those chemicals are being vetted and banned, and natural alternatives are now highly sought after. This study investigated for the first time some forest plants harvested in the southwest of China, which have not yet been explored as modulators of ruminant nutrition. This study provided information on the basic nutritional composition of the selected plants, the major phytochemicals, and the inhibitory effect on total rumen protozoa, ammoniagenesis, and α-glucosidase activity. 

The content of protein, NDF, ADF, and ash of *K. indica,*
*P. oleracea* and *B. rapa* subsp. *Chinensis* (tubers) was reported for the first time in this study. Abiona et al. [26] found a protein content of 13.6% in *A. digitata* leaves collected in the state of Oyo in Nigeria. *F. macrophylla* foliage collected from the southern part of Vietnam contained 16% protein, 64.7% NDF, and 53.4% ADF [27]. *A. digitata, K. indica, C. gigantea*, and *P. oleracea* had a protein content greater than 16%, and thus, they can be used as a potential source of protein for cattle. Moreover, fat-rich plants could be explored as sources of omega-3 and omega-6 fatty acids, which can promote health in both animals and humans [28]. Flavonoids are a group of phenolic compounds often found in plants, and they are highly regarded for their health-promoting and disease-preventing potential as well as antiviral and anti-protozoal activities [29,30]. Future studies are needed to evaluate if they affect palatability, feed intake, feed digestion, or microbiome in animal husbandry.

Several studies have recently reported the qualitative screening of phytochemicals including flavonols, terpenes, and cardenolide glycosides in *C. gigantea* [29,31,32]. A recent study also detected the presence of *p*-hydroxybenzoic acid, 4-O-*β*-d-galactopyranosyl-d-fructose, myrciacitrin IV, quinic acid, and derivatives of astragaloside/quinic acid in *C. procera*, a plant species of the same genus as *C. gigantea* [33]. *A. digitata* contained quercetin 3-*O*-glucoside [34], iridoid, phenylethanoid, and hydrocinnamic acid glycosides [35]. To the best of our knowledge, this is the first study that detected 3-hydroxybenzoic acid, astragaloside, myricetin, and quercetin in the tested plants. These bioactive compounds have demonstrated antioxidant, anti-inflammatory, antidiabetic [33], and antimicrobial activities, especially against pathogenic Gram-negative bacteria [36]. Moreover, antimicrobials found in the plants can be used to manipulate the rumen microbiota, thereby optimizing feed efficiency. 

The effect of *C. gigantea* on rumen protozoa at the genus level has been recently evaluated, and the *Entodinium* population, which dominates the rumen protozoal community, decreased by about 50% [4]. At the same time, the other genera of rumen protozoa, including *Isotricha, Epidinium*, and *Dasytricha*, were not inhibited, and thus, their contribution to fiber digestion should not be compromised. Moreover, holotrich protozoa (*Isotricha* and *Dasytricha*), which have a lower bacterivory activity compared with the entodiniomorphids [37], have a lower impact on NH_3_–N concentration [38] and duodenal microbial protein flow [39]. 

Studies on defaunated–refaunated animals using chemical agents, rumen washing, animal isolation, and immunological approaches have been previously carried out to suppress the growth of rumen protozoa [40,41]. However, none of those approaches are practical at the farm level [42]. Imidazole, a lysozyme inhibitor, inhibited rumen ciliate protozoa but did not decrease NH_3_–N production after 24h of in vitro incubation of protozoal cultures [43], which is consistent with our results observed with *C. gigantea*. We did note that *F. macrophylla* increased NH_3_–N production. Furthermore, myricetin and quercetin, previously reported as human synthetic inhibitors of total rumen protozoa counts [44], were also found in the tested plants. Future research can help confirm which of these two compounds is responsible for the observed inhibition of rumen protozoa by the plants.

The pellicles of *Entodinium* cells were damaged to different degrees by the tested plants. Specifically, the pellicles collapsed by all the tested plants but the normal longitudinal striations on the pellicle surface disappeared only after the exposure to *B. rapa* subsp. *chinensis*, *K. indica,* and *A. digitata*. The disruption of the pellicles surface structure of protozoa has been reported by Zeitz et al. [45] as one of the signs of the dying rumen ciliate protozoa. Such a disruption of the cell surface structure and cell death were also observed in *E. caudatum* cells exposed and then killed by some antibiotics [5]. It was believed that the destruction of cell surface structure leads to the loosened appearance of the filamentous glycocalyx, destruction of chromatin and granular nucleoli, and the accumulation of glycogen granules that finally obstructs the normal cell physical processes and ATP utilization [5,46]. Furthermore, the disruption of the protozoal cell surface could hinder the ectosymbionts associated with the protozoal cells, including methanogens [47]. Thus, the destruction of the protozoal pellicles could decrease CH_4_ production by methanogens associated with rumen protozoa. 

One of the well-recognized effective approaches to keep the serum level of glucose from rapidly rising is the inhibition of α-glucosidase with a specific inhibitor, such as acarbose [24]. Although this does not apply to ruminant animals because few dietary sugars can reach the small intestines, digestion of starch after the feeding of high-concentrate diets can lead to rapid release of glucose and subsequent fermentation to VFA and a decrease in rumen pH, potentially resulting in rumen acidosis. In lactating cows fed a high carbohydrate ration, the supplementation of acarbose prevented ruminal pH from lowering to the critical level inducing rumen acidosis [48]. Therefore, *A. digitata* and *B. rapa chinensis*, all of which can grow in dry or arid lands, may be used as alternatives to acarbose to lower the risk of or prevent rumen acidosis. We identified anthraquinone, 3-hydroxybenzoic acid, astragaloside, myricetin, and quercetin in the tested plants. Quercetin flavonoid has been shown to inhibit α-glucosidase and have hypoglycemic effects by improving and stabilizing the secretion and regeneration of insulin in human pancreatic islets with no significant adverse effect on health. The administration of myricetin in diabetic rats resulted in a 50% decrease in hyperglycemia and an augmentation in hepatic glycogen and glucose-6-phosphate content [49]. This clearly demonstrated that the plants tested in the present study could be explored as potential sources of anti-diabetic remedies. 

## 5. Conclusions

The present study demonstrated how new forest plants could be used as potential feed additives for animal husbandry. The tested plants contained bioactive compounds that could reduce the rumen ciliate protozoal population, potentially decreasing intra-ruminal protein recycling and thus improving dietary nitrogen utilization efficiency. Additionally, the cell surface of the protozoa was damaged, and the striation was collapsed, impairing the ectosymbiotic relationship with methanogens. In addition, *A. digitata* and *B. rapa chinensis* were able to inhibit the α-glucosidase enzyme, thereby potentially mitigating the risk of ruminal acidosis in high-producing cows fed high-concentrate diets. Overall, the leaves of the tested plants contain high levels of protein, especially the leaves of *A. digitata*, *K. indica**,*
*P. oleracea,* and *C. gigantea*, making them new sources of functional feed for ruminants. However, future studies are needed to evaluate other important nutritional traits, such as production of volatile fatty acids, digestibility, and palatability and to identify the anti-protozoal phytochemical(s) using bio-guided fractionation assays. 

## Figures and Tables

**Figure 1 biology-10-01055-f001:**
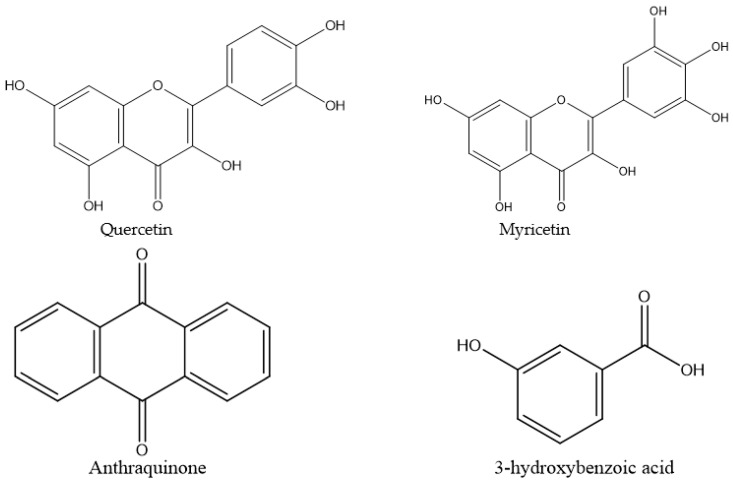
Chemical structure of the identified individual plant compounds. ChemDraw was used to generate the structure of the compounds based on their HPLC identification. The structure of astragaloside is not shown because it includes a group of compounds.

**Figure 2 biology-10-01055-f002:**
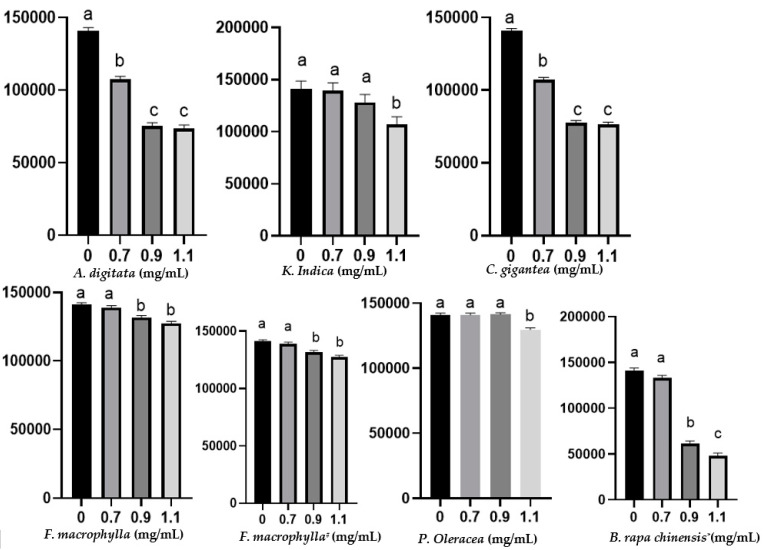
Total rumen protozoa count after 24 h of in vitro incubation at different doses of each plant. The Y-axis represents the total number of protozoa cells per mL of culture fluid. Different superscripts on the bars within each graph denote significant difference (*p* < 0.05) among plant doses. * Roots were used; # tubers were used.

**Figure 3 biology-10-01055-f003:**
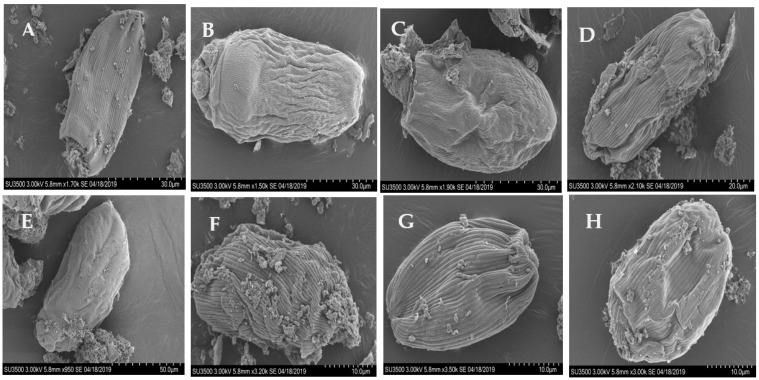
Scanning electron micrographs of *Entodinium* cells in the in vitro cultures after 24 h of incubation. (**A**) control, no plant inclusion, (**B**) *Brassica rapa* subsp. *chinensis*, (**C**) *Kalimeris indica*, (**D**) *Calotropis gigantea*, (**E**) *Adansonia digitata*, (**F**) *Portulaca oleracea*, (**G**) *Flemingia macrophylla* (roots), and (**H**) *Flemingia macrophylla* (leaves). The cells surface changes were observed at the highest tested dose (1.1 mg/mL) for each plant. No morphological changes to the cell surfaces were observed at the two lower doses or the negative control (not shown).

**Figure 4 biology-10-01055-f004:**
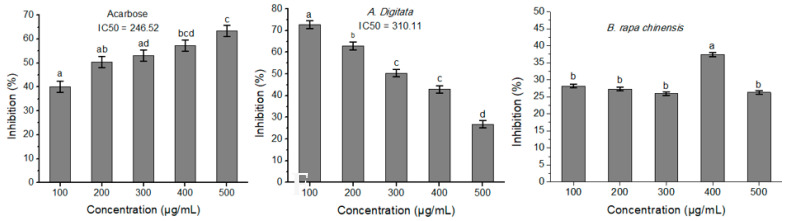
Inhibition of α-glucosidase activities by the extract of *A. digitata, B. rapa chinensis* leaves, and acarbose. *B. rapa chinensis* extract did not show significant inhibition at the tested concentrations. The extracts of the other plants showed unexpected α-glucosidase inhibition and warrant further analysis (Appendix A). Different superscripts within each graph denote significant difference (*p* < 0.05) among plant extract concentrations.

**Table 1 biology-10-01055-t001:** Nutritional evaluation and total polyphenol composition of the tested plants.

Plants Species	Nutrient Content (%, Mean ± SD)						Total Polyphenols			
	DM	CP	EE	NDF	ADF	Ash	CEY (%)	TPC (µg/g)	TFC (µg/g)	TAC (µg/g)
*A. digitata*	21.5 ± 0.1 ^e^	19.8 ± 0.2 ^b^	1.9 ± 0.0 ^bc^	29.7 ± 0.9 ^c^	24.2 ± 1.0 ^e^	10.5 ± 0.2 ^de^	15.1 ± 0.5 ^c^	5947.0 ± 0.6 ^c^	715.2 ± 0.6 ^c^	410.6 ± 0.6 ^e^
*K. indica*	22.3 ± 0.2 ^g^	28.5 ± 0.3 ^a^	1.8 ± 0.2 ^c^	53.2 ± 0.6 ^c^	45 ± 0.1 ^d^	12.8 ± 0.3 ^bc^	19.1 ± 0.5 ^b^	2766.7 ± 0.6 ^e^	580.0 ±0.6 ^d^	253.3 ± 0.6 ^f^
*C. gigantea*	25.5 ± 0.1 ^f^	17.1 ± 0.2 ^c^	3.9 ± 0.1 ^a^	30.7 ± 2.1 ^b^	54.4 ± 0.6 ^c^	16.1 ± 0.2 ^a^	13.5 ± 0.5 ^c^	3266.7 ± 0.6 ^d^	1866.7 ± 0.6 ^a^	440.0 ± 0.6 ^d^
*F. macrophylla*	31.9 ± 0.1 ^d^	10.5 ± 0.1 ^d^	0.9 ± 0.1 ^e^	49.0 ± 1.8 ^ab^	34.4 ± 1.0 ^b^	8.5 ± 1.4 ^f^	9.4 ± 0.5 ^d^	15,466.6 ± 0.6 ^a^	746.7 ± 0.6 ^b^	906.7 ± 0.6 ^a^
*F.macrophylla ^#^*	38.9 ± 0.2 ^b^	7.3 ± 0.2 ^e^	0.2 ± 0.1 ^f^	53.1 ± 1.8 ^a^	29.3 ± 1.0 ^a^	3.3 ± 0.1 ^g^	13.7 ± 0.5 ^c^	9166.0 ± 0.6 ^b^	546.7 ± 0.6 ^e^	840.0 ± 0.6 ^b^
*P. oleracea*	32.4 ± 0.3 ^c^	16.6 ± 0.1 ^c^	1.4 ± 0.1 ^d^	59.3 ± 2.1 ^d^	19 ± 1 ^f^	9.5 ± 0.5 ^ef^	8.1 ± 0.5 ^d^	1413.3 ± 0.6 ^f^	540.0 ± 0.6 ^f^	133.3 ± 0.6 ^g^
*B. rapa c. **	29.4 ± 0.1 ^a^	10.0 ± 0.3 ^d^	0.4 ± 0.0 ^f^	20.7 ± 0.8 ^cd^	20 ± 0.8 ^f^	11.2 ± 0.1 ^cd^	40.4 ± 0.7 ^e^	1106.7 ± 0.7 ^g^	520.8 ± 0.7 ^g^	485.5 ± 0.6 ^c^

DM: dried matter, CP: crude protein, EE: ether extract, NDF: neutral detergent fiber, ADF: acid detergent fiber, CEY: crude extract yield, TPC: total phenolic compounds, TFC: total flavonoid compounds, TAC: total alkaloid content. Different superscripts (a, b, c, d, e, f, and g) within a column denote significant difference (*p* < 0.05) among plants. ^#^ Only roots were used; * only tubers were used.

**Table 2 biology-10-01055-t002:** Phytochemical compounds found in the extract of the forest plants.

Name	Formula	MW	RT (min)	Concentration (µg/g Plant)							SEM	*p*-Value
				*A. digitata*	*K. indica*	*C. gigantea*	*F. macrophylla*	*F. macrophylla* ^#^	*P. oleracea*	*B. rapa* *		
Quercetin	C_16_H_10_O_7_	302.24	15.34	157.53 ^b^	NF	5884.29 ^a^	NF	101.79 ^c^	96.17 ^c^	76.39 ^d^	3.34	0.001
Anthraquinone	C_14_H_8_O_2_	208.22	21.39	82.91 ^c^	NF	1121.77 ^a^	NF	100.20 ^b^	NF	NF	2.89	0.001
3-HOBA	C_7_H_6_O_3_	138.12	8.88	17,248.73 ^b^	40,026.80 ^a^	1020.31 ^d^	829.06 ^e^	4514.97 ^c^	283.02 ^f^	256.99 ^g^	3.09	0.001
Astragaloside	C_28_H_32_O_17_	NF	19.23	14,115.36 ^a^	NF	NF	11,521.21 ^d^	11,580.77 ^c^	11,177.09 ^e^	13,427.11 ^b^	2.89	0.001
Myricetin	C_21_H_20_O_12_	464.38	11.93	4756.89 ^a^	649.00 ^b^	451.31 ^c^	296.40 ^d^	237.58 ^e^	98.21 ^f^	103.892 ^f^	3.67	0.001

MW: molecular weight, kDa; RT: retention time in minutes; NF: not found in the plant; ***^#^*** only roots were used; *: only tubers were used; 3-HOBA: 3-hydroxibenzoic acid; SEM: standard error of the mean; different superscripts (a, b, c, d, and f) within a row denote significant difference (*p* < 0.05) among plants.

**Table 3 biology-10-01055-t003:** Ammonia nitrogen (NH_3_–N) concentration of the in vitro cultures after 24 h incubation.

Plant Doses (mg/mL)	N-NH_3_ (mg/dL)				SEM	Trt	*p*-Value	
	0	0.7	0.9	1.1			Linear	Quadratic
*A. digitata*	27.4	27.0	27.4	27.7	0.8	0.990	0.941	0.778
*K.indica*	27.4	26.3	26.0	27.3	1.2	0.872	0.744	0.521
*C. gigantea*	27.4 ^a^	26.7 ^a^	26.5 ^a^	18.5 ^b^	0.8	0.004	0.006	0.007
*F. macrophylla*	27.4 ^b^	27.0 ^b^	27.2 ^b^	32.2 ^a^	0.6	0.021	0.053	0.013
*F.macrophylla ^#^*	27.4 ^b^	27.2 ^b^	27.3 ^b^	32.6 ^a^	0.7	0.034	0.066	0.023
*P. oleracea*	27.4	27.0	27.1	27.7	1.8	0.990	0.941	0.778
*B. rapa chinensis **	27.4	27.4	31.4	30.7	0.6	0.056	0.023	0.248

SEM: standard error the mean, Trt: treatment, ^#^ only roots were used, * only tubers were used, different superscripts within a row denote significant difference (*p* < 0.05) among plant doses.

## Data Availability

Not applicable.

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
