# Peer review of "Identification of Bioactive Phytochemicals from Six Plants: Mechanistic Insights into the Inhibition of Rumen Protozoa, Ammoniagenesis, and α-Glucosidase"

_biology, 2021, doi:10.3390/biology10101055_

Round 1
Reviewer 1 Report
Title and general comments: Identification of bioactive phytochemicals from six plants: In- 2 sights into the mechanistic inhibition of rumen protozoa, am- 3 moniagenesis and α-glucosidase.
It is an interesting study with some new findings. However, the manuscript could be improved a bit more with some additional data.
Comment 1: "Mechanistic disruption of the extracellular structure of the protozoa, which might be contributable to the observed inhibition" it looks like an assumption, is there any way to exactly show the reasons for the inhibiting activity?.
Comment 2: Please have a thorough language check and mark the changes in the revised manuscript. If possible, have a thorough proofread with the help of a native speaker.
Comment 3: Please mention more detailed information on rumen fluid collection and animals.
Comment 4: In line 154, its mentioned animals were handled carefully, but I couldn't find an ethical approval number. Would you please mention it?
Comment 5: Please check the figure clarity, especially in figure 1 on structure names.
Author Response
It is an interesting study with some new findings. However, the manuscript could be improved a bit more with some additional data.
Comment 1: "Mechanistic disruption of the extracellular structure of the protozoa, which might be contributable to the observed inhibition" it looks like an assumption, is there any way to exactly show the reasons for the inhibiting activity?.
AU: In a previous study, the destruction of the extracellular structure was shown to link to the death of Entodinium caudatum, the most common species of rumen protozoa. That study is now cited in the revised manuscript (lines 440-442 of the Discussion).
Comment 2: Please have a thorough language check and mark the changes in the revised manuscript. If possible, have a thorough proofread with the help of a native speaker.
AU: The manuscript has been edited and proofread and the track changes are marked.
Comment 3: Please mention more detailed information on rumen fluid collection and animals.
AU: More details on rumen fluid collection and animals have been added (lines 185-195).
Comment 4: In line 154, its mentioned animals were handled carefully, but I couldn't find an ethical approval number. Would you please mention it?
AU: The ethical approval number has been added (No. IAS20180115) (line 182)
Comment 5: Please check the figure clarity, especially in figure 1 on structure names.
AU: The figure clarity has been improved (lines 307-309)

Reviewer 2 Report
In the current study, the authors aimed to evaluate certain plants for their inhibitory effect on total rumen protozoa and α-glucosidase and identifying their major phytochemicals to aim mechanistic understanding of their effects. This study is interesting and may represent novel data but certain points should be considered before publication.
Major points:
1- In abstract, the authors should mention the plant names used in the study before the results. Also, the graphical abstract should include the names of plants rather than writing six plants. The graphical abstract needs more improvement to be self representative.
2- α-glucosidase is mentioned as a keyword but it was not mentioned in the abstract section to indicate its role in the study.
3- It is not clear which are the six plants. For example, table only includes 5 plants in addition to A. digitata. F. macrophylla was mentioned two times, one time as leaves and one time as roots. In line 94 in introduction and everywhere in the manuscript, authors mentioned that this study aimed to analyze another six plants in comparison with C. gigante. Actually, the current data represent only five plants while it should provide data regarding 6 plants other than C. gigantea as I understood.
4- Title of table one should not be restricted to plants leaves because data include roots and tubers in certain plants.
5- Section 3.4 mentioned % inhibition of protozoal inhibition for each plant but the figure 2 did not represent any % inhibition on the Y-axis. Also, the authors should indicate in methods section how they calculate % inhibition. Where is the title for Y-axis?
6- What is the meaning of a, b,c symbols for statistics in each figure? It was not mentioned in the manuscript and in figure legends.
7- Figure 4 did not represent the effect of all plants and the authors mentioned that they showed strange data. However, the data should be supplied even as supplementary data.
8- Why did not authors analyze the types of protozoa and percent inhibition induced by each plant? Lacking of sufficient biological data and detailed mechanisms of action makes this manuscript highly related to journals of natural products research rather than biology journal.
Minor points:
1- In keywords: please, remove (;) after protozoa inhibition.
2- Figure 1: the font of compound names should be the same size and style.
3- Manuscript title should be improved.
Author Response
In the current study, the authors aimed to evaluate certain plants for their inhibitory effect on total rumen protozoa and α-glucosidase and identifying their major phytochemicals to aim mechanistic understanding of their effects. This study is interesting and may represent novel data but certain points should be considered before publication.
Major points:
1- In abstract, the authors should mention the plant names used in the study before the results. Also, the graphical abstract should include the names of plants rather than writing six plants. The graphical abstract needs more improvement to be self representative.
AU: The names of the plants have been added in the abstract and on the graphical abstract. More information was added to the graphic abstract (lines 40-44; line 66).
2- α-glucosidase is mentioned as a keyword but it was not mentioned in the abstract section to indicate its role in the study.
AU: Due to the word limit of the abstract, we could not define the role of α-glucosidase here. It is elucidated in the introduction part.
3- It is not clear which are the six plants. For example, table only includes 5 plants in addition to A. digitata. F. macrophylla was mentioned two times, one time as leaves and one time as roots. In line 94 in introduction and everywhere in the manuscript, authors mentioned that this study aimed to analyze another six plants in comparison with C. gigante. Actually, the current data represent only five plants while it should provide data regarding 6 plants other than C. gigantea as I understood.
AU: We agree, the sentence has been modified to: ‘The objective of the present study was to analyze another five plants in comparison with C. gigantea (line 119)
4- Title of table one should not be restricted to plants leaves because data include roots and tubers in certain plants.
AU: We agree, ‘’plant leaves’’ has been replaced by ‘’tested plants’’ (line 297)
5- Section 3.4 mentioned % inhibition of protozoal inhibition for each plant but the figure 2 did not represent any % inhibition on the Y-axis. Also, the authors should indicate in methods section how they calculate % inhibition. Where is the title for Y-axis?
AU: The calculation method of % inhibition has been inserted (lines 215-218).
Inhibition (%) = Ndi x 100 / Ncontrol
Where Ndi is the number of cells at the dose i (0.7, 0.9 or 1.1 mg/mL) and Ncontrol is the number of cells without supplementing any plant.
We explained the Y-axis in the legend of figure 2. We tried to add it directly on the figure but that made the figure more repetitive and crowded (lines 340-341).
6- What is the meaning of a, b,c symbols for statistics in each figure? It was not mentioned in the manuscript and in figure legends.
AU: The statistical meaning of a, b, c in each figure has been explained in the legends (line 300).
7- Figure 4 did not represent the effect of all plants and the authors mentioned that they showed strange data. However, the data should be supplied even as supplementary data.
AU: A supplementary figure has been added at the end of the manuscript to show the data including the unexpected results of some plants (lines 625-633)
8- Why did not authors analyze the types of protozoa and percent inhibition induced by each plant? Lacking of sufficient biological data and detailed mechanisms of action makes this manuscript highly related to journals of natural products research rather than biology journal.
AU: In this study we did not determine the inhibition to individual taxa (or genera) of rumen protozoa for two reasons. First, Entodinium is the predominant rumen protozoa, accounting for more than 95% of the total rumen protozoa in domesticated ruminants. Second, the focus of this study was to screen the tested plants to provide the basis for our next research that will consider not only the analysis of protozoa but also the other microbiota such as bacteria and archaea.
In this study we aimed to identify the phytochemicals potentially responsible for the inhibition of rumen protozoa and gain some mechanistic understanding of their inhibition. Thus, we think this study is within the scope of this Biology, which is an interdisciplinary journal. We described and discussed the mechanisms of action of the protozoa inhibition and the biological implication on section 3.5 of the Results (lines 325-328) and at the 6th paragraph of the Discussion section (lines 436-449) (The pellicles of Entodinium cells were damaged to different degrees by the tested plants. Specifically, the pellicles collapsed by all the tested plants but the normal longitudinal striations on the pellicle surface disappeared only after the exposure to B. rapa subsp. chinensis, K. indica, and A. digitata. The disruption of the pellicles surface structure of protozoa has been reported by Zeitz et al. [38] as one of the signs of the dying rumen ciliate protozoa. Such a disruption of the cell surface structure leads to the loosened appearance of the filamentous glycocalyx, destruction of chromatin and granular nucleoli, the accumulation of glycogen granules that finally obstructs the normal cell physical processes and ATP utilization [5, 39]. Furthermore, the disruption of the protozoal cell surface could hinder the ectosymbionts associated with the protozoal cells, including methanogens [40]. Thus, the destruction of the protozoal pellicles could decrease CH4 production by methanogens associated with rumen protozoa).
Minor points:
1- In keywords: please, remove (;) after protozoa inhibition.
AU: ; has been removed (line 63)
2- Figure 1: the font of compound names should be the same size and style.
AU: The figure has been amended (lines 306-308)
3- Manuscript title should be improved.
AU: the title is modified to make it more clear and reflective of the study (line 2-3).

Reviewer 3 Report
The manuscript (Manuscript ID biology-1362845) by Ayemele et al., is a research article which deals with the identification of bioactive phytochemicals from six plants. It was very interesting to read the complete manuscript. After reading and thoroughly analyzing the text, data (figures, and tables) presented by the authors, I have decided to accept this manuscript after major revision. I have attempted to mention some of the important points to improve the quality of the paper for future submission. Please have a look at some of the comments/suggestions for revision:
- The standard of the English language is variable among different sections of the manuscript. Authors are suggested to check the standard of the English language to make the manuscript free of any grammar/scientific or typographic errors.
- Theme fonts size is different at various places in the manuscript. Please recheck it and make it similar throughout the manuscript.
- In the abstract, the authors should add a statement, in the end highlighting the aim/coverage and significance of the current findings. Also, mention the limitation of the present study.
- The introduction and other sections of the review should be improved and should be up to date. It should include more recent trends, currently known research findings. Currently, a large number of studies have shown that plants contain different bioactive phytochemicals which are being used in various therapeutic settings. The authors should elaborate on this in more detail. In this context, the following references would be helpful for a better understanding of the background section.
https://www.mdpi.com/2218-1989/9/11/258
- Mechanisms are not written well. This decreases the scientific interest and value.
- ‘Discussion and Conclusion sections’ are not well represented based on the theme, data presented, in the manuscript. The authors should elaborate on this section in more detail.
- It is very important to mention the future scope/direction of the present study at the end of the conclusion.
- Based on the above comments and suggestions I would recommend this manuscript as a major revision. It will be a pleasure to read the revised manuscript.
Author Response
The manuscript (Manuscript ID biology-1362845) by Ayemele et al., is a research article which deals with the identification of bioactive phytochemicals from six plants. It was very interesting to read the complete manuscript. After reading and thoroughly analyzing the text, data (figures, and tables) presented by the authors, I have decided to accept this manuscript after major revision. I have attempted to mention some of the important points to improve the quality of the paper for future submission. Please have a look at some of the comments/suggestions for revision:
- The standard of the English language is variable among different sections of the manuscript. Authors are suggested to check the standard of the English language to make the manuscript free of any grammar/scientific or typographic errors.
AU: Thank you for the suggestion. The manuscript has been revised, edited, and proofread
- Theme fonts size is different at various places in the manuscript. Please recheck it and make it similar throughout the manuscript.
AU: Apology for this. We made the font consistent throughout the manuscript.
- In the abstract, the authors should add a statement, in the end highlighting the aim/coverage and significance of the current findings. Also, mention the limitation of the present study.
AU: The significance of the current findings as well as the limitations have been made more explicit in the abstract (lines 57-61)
The aims of the study are mentioned in the simple summary per the journal’s guidelines. The objective is also added in the abstract (lines 40-44)
- The introduction and other sections of the review should be improved and should be up to date. It should include more recent trends, currently known research findings. Currently, a large number of studies have shown that plants contain different bioactive phytochemicals which are being used in various therapeutic settings. The authors should elaborate on this in more detail. In this context, the following references would be helpful for a better understanding of the background section.
https://www.mdpi.com/2218-1989/9/11/258
AU: Thank you for sharing the reference that helped to add more of the current research findings on phytochemicals used in therapeutic settings. We revised the Introduction to include more relevant background information to strengthen the justification of this study (lines 104-113)
- Mechanisms are not written well. This decreases the scientific interest and value.
AU: We described and discussed the mechanisms of action of the protozoa inhibition and the biological implication in section 3.5 of the Results and the 6th paragraph of the Discussion section. (lines 325-328) and (lines 436-449)
- ‘Discussion and Conclusion sections’ are not well represented based on the theme, data presented, in the manuscript. The authors should elaborate on this section in more detail.
AU: In the conclusion part, more mechanism of protozoa inhibition has been added (line 471-473)
In the discussion part, we mentioned that except for A. digitata, the phytochemicals analysis of the tested plants were evaluated for the first time in this study and we discussed some of the biological functions of the identified phytochemicals, in relation with our theme. (lines 392-397)
Also, the protozoa inhibition has been discussed in comparison with the antimicrobials chemical inhibitors previously reported, including the morphological changes after exposure to the inhibitors. Lines (398-435)
Moreover, the biological implications of the inhibition on the intra-cellular structure of protozoa, ATP metabolism and microbial ectosymbiosis have been discussed. Finally, the inhibition of ammoniagenesis and a-glucosidase is also discussed (lines 436-449).
- It is very important to mention the future scope/direction of the present study at the end of the conclusion.
AU: Yes indeed. The future direction of the study was mentioned at the end of the conclusion (lines 478-481)
- Based on the above comments and suggestions I would recommend this manuscript as a major revision. It will be a pleasure to read the revised manuscript.
AU: thank you for your very constructive comments and suggestions, which helped us in improving this manuscript.

Round 2
Reviewer 2 Report
The authors covered most of the points. These two points still need to be covered:
1- Figure 2 represents only total rumen protozoal count while section 3.4 compared inhibition % between plants. The figure should represent what indicated in the text. The figure should be changed to represent inhibiton % or the reverse. (as mentioned in major point no. 5 in the previous report).
2- In major point 6 in the previous report, authors mentioned that the statistical meaning of a,b, c in each figure has been explained in the legends (line 300):
but sorry I cannot find any explanation for these symbols. Most line numbers indicated in the reply are not updated to the revised version.
Author Response
Comments and Suggestions for Authors
In the current study, the authors aimed to evaluate certain plants for their inhibitory effect on total rumen
protozoa and α-glucosidase and identifying their major phytochemicals to aim mechanistic understanding
of their effects. This study is interesting and may represent novel data but certain points should be
considered before publication.
Major points:
1- In abstract, the authors should mention the plant names used in the study before the results. Also, the
graphical abstract should include the names of plants rather than writing six plants. The graphical abstract
needs more improvement to be self representative.
AU: The names of the plants have been added in the abstract and on the graphical abstract. More
information was added to the graphic abstract (lines 38-39; line 56).
2- α-glucosidase is mentioned as a keyword but it was not mentioned in the abstract section to indicate its
role in the study.
AU: Due to the word limit of the abstract, we could not define the role of α-glucosidase here. It is
elucidated in the introduction part.
3- It is not clear which are the six plants. For example, table only includes 5 plants in addition to A.
digitata. F. macrophylla was mentioned two times, one time as leaves and one time as roots. In line 94 in
introduction and everywhere in the manuscript, authors mentioned that this study aimed to analyze
another six plants in comparison with C. gigante. Actually, the current data represent only five plants
while it should provide data regarding 6 plants other than C. gigantea as I understood.
AU: We agree, the sentence has been modified to: ‘The objective of the present study was to analyze
another five plants in comparison with C. gigantea (line 110)
4- Title of table one should not be restricted to plants leaves because data include roots and tubers in
certain plants.
AU: We agree, ‘’plant leaves’’ has been replaced by ‘’tested plants’’ (line 285)
5- Section 3.4 mentioned % inhibition of protozoal inhibition for each plant but the figure 2 did not
represent any % inhibition on the Y-axis. Also, the authors should indicate in methods section how they
calculate % inhibition. Where is the title for Y-axis?
AU: The proportions presented in the text is now changed into the number of cells (lines 300-304) to
make it consistent with the figure (At the tested maximum dose of 1.1 mg/mL, B. rapa subsp. chinensis
had a higher inhibitory effect, decreasing the protozoa counts from 140000 to 48000 cells/mL while A.
digitata and C. gigantea reduced protozoa from 140000 to 73360 and to 76160 respectively (P< 0.05). K.
indica, F. macrophylla, and P. oleracea reduced the protozoal count to 106400, 126700, and 128800,
respectively).
We explained the Y-axis in the legend of figure 2. We tried to add it directly on the figure but that made the figure
more repetitive and crowded (line 324).
6- What is the meaning of a, b,c symbols for statistics in each figure? It was not mentioned in the
manuscript and in figure legends.
AU: The statistical meaning of superscripts in each figure has been explained in the legends (line 288).
(Means with different superscripts (a, b, c, d, e, f, g) within a column 288 differ (P < 0.05))
7- Figure 4 did not represent the effect of all plants and the authors mentioned that they showed strange
data. However, the data should be supplied even as supplementary data.
AU: A supplementary figure has been added at the end of the manuscript to show the data including the
unexpected results of some plants (lines 602-609)
8- Why did not authors analyze the types of protozoa and percent inhibition induced by each plant?
Lacking of sufficient biological data and detailed mechanisms of action makes this manuscript highly
related to journals of natural products research rather than biology journal.
AU: In this study we did not determine the inhibition to individual taxa (or genera) of rumen protozoa
for two reasons. First, Entodinium is the predominant rumen protozoa, accounting for more than 95% of
the total rumen protozoa in domesticated ruminants. Second, the focus of this study was to screen the
tested plants to provide the basis for our next research that will consider not only the analysis of
protozoa but also the other microbiota such as bacteria and archaea.
In this study we aimed to identify the phytochemicals potentially responsible for the inhibition of rumen
protozoa and gain some mechanistic understanding of their inhibition. Thus, we think this study is within
the scope of this Biology, which is an interdisciplinary journal. We described and discussed the
mechanisms of action of the protozoa inhibition and the biological implication on section 3.5 of the
Results (lines 310-313) and at the 6th paragraph of the Discussion section (Lines 416-429)
Lines 310-313: The extracellular surface (pellicles) of Entodinium cells collapsed and wilted with the
tested plants (Figure 3). The normal longitudinal striations present on the cell surface disappeared after
the exposure to B. rapa subsp. chinensis, K. indica, and A. digitata, but C. gigantea, F. macrophylla, P.
oleracea, or the control did not destruct the striations
Lines 416-429: The pellicles of Entodinium cells were damaged to different degrees by the tested plants.
Specifically, the pellicles collapsed by all the tested plants but the normal longitudinal striations on the
pellicle surface disappeared only after the exposure to B. rapa subsp. chinensis, K. indica, and A. digitata.
The disruption of the pellicles surface structure of protozoa has been reported by Zeitz et al. [38] as one
of the signs of the dying rumen ciliate protozoa. Such a disruption of the cell surface structure leads to
the loosened appearance of the filamentous glycocalyx, destruction of chromatin and granular nucleoli,
the accumulation of glycogen granules that finally obstructs the normal cell physical processes and ATP
utilization [5, 39]. Furthermore, the disruption of the protozoal cell surface could hinder the
ectosymbionts associated with the protozoal cells, including methanogens [40]. Thus, the destruction of
the protozoal pellicles could decrease CH4 production by methanogens associated with rumen
protozoa). Minor points:
1- In keywords: please, remove (;) after protozoa inhibition.
AU: ; has been removed (line 54)
2- Figure 1: the font of compound names should be the same size and style.
AU: The figure has been amended (lines 290-293)
3- Manuscript title should be improved.
AU: the title is modified to make it more clear and reflective of the study (line 2-4).

Reviewer 3 Report
- Authors have not reported the mode of action in this study. Mode of action must be carried out.
- Authors should represent the physical properties of the screened compounds.
- Microbiome analysis is missing. Authors must represent this as it is very crucial.
- Authors should clearly represent the effect of studied enzymes on the microbiome.
Author Response
Reviewer 3
The manuscript (Manuscript ID biology-1362845) by Ayemele et al., is a research article which deals with
the identification of bioactive phytochemicals from six plants. It was very interesting to read the complete
manuscript. After reading and thoroughly analyzing the text, data (figures, and tables) presented by the
authors, I have decided to accept this manuscript after major revision. I have attempted to mention
some of the important points to improve the quality of the paper for future submission. Please have a look
at some of the comments/suggestions for revision:
1. The standard of the English language is variable among different sections of the manuscript.
Authors are suggested to check the standard of the English language to make the manuscript
free of any grammar/scientific or typographic errors.
AU: Thank you for the suggestion. The manuscript has been revised, edited, and proofread
2. Theme fonts size is different at various places in the manuscript. Please recheck it and make it
similar throughout the manuscript.
AU: Apology for this. We made the font consistent throughout the manuscript.
3. In the abstract, the authors should add a statement, in the end highlighting the aim/coverage and
significance of the current findings. Also, mention the limitation of the present study.
AU: The significance of the current findings as well as the limitations have been made more explicit in the
abstract (lines 49-50)
The aims of the study are mentioned in the simple summary per the journal’s guidelines. Meanwhile we also added the objective in the abstract (lines 37-41)
4. The introduction and other sections of the review should be improved and should be up to date. It
should include more recent trends, currently known research findings. Currently, a large number
of studies have shown that plants contain different bioactive phytochemicals which are being
used in various therapeutic settings. The authors should elaborate on this in more detail. In this
context, the following references would be helpful for a better understanding of the background
section.
https://www.mdpi.com/2218-1989/9/11/258
AU: Thank you for sharing the reference that helped to add more of the current research findings on
phytochemicals used in therapeutic settings. We revised the Introduction to include more relevant
background information to strengthen the justification of this study (lines 95-104)
Natural plants are a potential source of novel biologically active compounds that 95 could lead to
metabolic interventions as new therapeutics [9,10]. The recent discovery of 96 a powerful antimalarial
drug, artemisinin, derived from Artemisia annua L. (sweet worm- 97 wood) [11] is recommended by the
World Health Organization to cure malaria disease, 98 caused by Plasmodium falciparum parasite [12].
Traditional Chinese medicine is devoted to 99 treating a wide range of infectious diseases [13], and
several phytochemical compounds are already undergoing clinical trials [14]. High throughput screening
has enabled a variety of phytochemicals to be searched for their chemical, bioactivity, and ethnobotany
information (https://phytochem.nal.usda.gov/phytochem/search/list) and for their potential targets of
proteins and nucleic acid targets (https://db.idrblab.org/ttd/) [15].
5. Mechanisms are not written well. This decreases the scientific interest and value.
AU: We described and discussed the mechanisms of action of the protozoa inhibition and the biological
implication in section 3.5 (Lines 310-313) of Results and the 6th paragraph of the Discussion sections.
Lines 310-313: The extracellular surface (pellicles) of Entodinium cells collapsed and wilted with the
tested plants (Figure 3). The normal longitudinal striations present on the cell surface disappeared after
the exposure to B. rapa subsp. chinensis, K. indica, and A. digitata, but C. gigantea, F. macrophylla, P.
oleracea, or the control did not destruct the striations
Lines 416-429: The pellicles of Entodinium cells were damaged to different degrees by the tested plants.
Specifically, the pellicles collapsed by all the tested plants but the normal longitudinal striations on the
pellicle surface disappeared only after the exposure to B. rapa subsp. chinensis, K. indica, and A. digitata.
The disruption of the pellicles surface structure of protozoa has been reported by Zeitz et al. [38] as one
of the signs of the dying rumen ciliate protozoa. Such a disruption of the cell surface structure leads to
the loosened appearance of the filamentous glycocalyx, destruction of chromatin and granular nucleoli,
the accumulation of glycogen granules that finally obstructs the normal cell physical processes and ATP
utilization [5, 39]. Furthermore, the disruption of the protozoal cell surface could hinder the
ectosymbionts associated with the protozoal cells, including methanogens [40]. Thus, the destruction of
the protozoal pellicles could decrease CH4 production by methanogens associated with rumen
protozoa).
6. ‘Discussion and Conclusion sections’ are not well represented based on the theme, data
presented, in the manuscript. The authors should elaborate on this section in more detail.
AU: In the conclusion part, more mechanism of protozoa inhibition has been added (line 450-453)
In the discussion part, we mentioned that except for A. digitata, the phytochemicals analysis of the tested
plants were evaluated for the first time in this study and we discussed some of the biological functions of
the identified phytochemicals, in relation with our theme. (lines 374-397)
Also, the protozoa inhibition has been discussed in comparison with the antimicrobials chemical inhibitors
previously reported, including the morphological changes after exposure to the inhibitors. Lines (421-422)
Moreover, the biological implications of the inhibition on the intra-cellular structure of protozoa, ATP
metabolism and microbial ectosymbiosis have been discussed (422-429).
Finally, the inhibition of ammoniagenesis (NH3N) and a-glucosidase is also discussed respectively on
(lines 409-411) and (line 430-446).
7. It is very important to mention the future scope/direction of the present study at the end of the
conclusion.
AU: Yes indeed. The future direction of the study was mentioned at the end of the conclusion (lines 458-
460): However, future studies are needed to evaluate other important nutritional 458 traits, such as
production of volatile fatty acids, digestibility and palatability, and to identify the anti-protozoal
phytochemical(s) using bio-guided fractionation assays
8. Based on the above comments and suggestions I would recommend this manuscript as a major
revision. It will be a pleasure to read the revised manuscript.
AU: thank you for your very constructive comments and suggestions, which helped us in improving
this manuscript.

Round 3
Reviewer 2 Report
The second point in my second report has not been covered yet. What are differences between symbols of signficance a, b,c ,d ,..? What are they mean? are they related to comparison to different groups? It is very important to indicate the group for each symbol ot what are the differences between these sybmols.
Author Response
The second point in my second report has not been covered yet. What are differences between symbols of signficance a, b,c ,d ,..? What are they mean? are they related to comparison to different groups? It is very important to indicate the group for each symbol ot what are the differences between these sybmols.
AU: The sentence has been changed to ''Different superscripts (a, b, c, d, e, f, g) within a column denote significant difference (P < 0.05) among plants''
The same modification is made for all the table/figure footnotes when necessary.
Thank you for your comment/input that helped to improve the quality of this manuscript.
Reviewer 3 Report
The Manuscript looks ok to me.
Author Response
The Manuscript looks ok to me.
AU: Thank you for your comments/inputs that helped to improve the quality of this manuscript.